Comparative genomic analysis of the PKS genes in five species and expression analysis in upland cotton

Su Xueqiang
Sun Xu
Cheng Xi
Wang Yanan
Abdullah Muhammad
Li Manli
Li Dahui
Gao Junshan
Cai Yongping 1806149539@QQ.COM ypcaiah@163.com
Lin Yi linyi992547404@163.com
School of Life Science, Anhui Agricultural University , Hefei , China
McCormick Sheila
Electronic publication date: 2017 Oct 30
Publication date: 2017
Volume: 5
Electronic Location ID: e3974
Received 2017 May 29; Accepted 2017 Oct 10
Copyright: ©2017 Su et al.
Copyright year: 2017
Copyright holder: Su et al.
License: This is an open access article distributed under the terms of the Creative Commons Attribution License, which permits unrestricted use, distribution, reproduction and adaptation in any medium and for any purpose provided that it is properly attributed. For attribution, the original author(s), title, publication source (PeerJ) and either DOI or URL of the article must be cited.
License URL: https://creativecommons.org/licenses/by/4.0/

Keywords: Upland cotton, Polyketide synthase, Procyanidins, Gene expression, Genome-wide analysis

Funding: National Natural Science Foundation of China 31640068 Cotton Industry Innovation System of Anhui Province Natural Science Foundation of the Higher Education Institutions of Anhui Province KJ2016A840 KJ2014A078 Postgraduate Student’s Innovation Fund of Anhui Agricultural University 2017yjs-32 This work was supported by grants from the National Natural Science Foundation of China (No. 31640068), the Cotton Industry Innovation System of Anhui Province, the Natural Science Foundation of the Higher Education Institutions of Anhui Province (No. KJ2016A840 and KJ2014A078) and Postgraduate Student’s Innovation Fund of Anhui Agricultural University (No. 2017yjs-32). The funders had no role in study design, data collection and analysis, decision to publish, or preparation of the manuscript.

==============================
Plant type III polyketide synthase (PKS) can catalyse the formation of a series of secondary metabolites with different structures and different biological functions; the enzyme plays an important role in plant growth, development and resistance to stress. At present, the PKS gene has been identified and studied in a variety of plants. Here, we identified 11 PKS genes from upland cotton (Gossypium hirsutum) and compared them with 41 PKS genes in Populus tremula, Vitis vinifera, Malus domestica and Arabidopsis thaliana. According to the phylogenetic tree, a total of 52 PKS genes can be divided into four subfamilies (I–IV). The analysis of gene structures and conserved motifs revealed that most of the PKS genes were composed of two exons and one intron and there are two characteristic conserved domains (Chal_sti_synt_N and Chal_sti_synt_C) of the PKS gene family. In our study of the five species, gene duplication was found in addition to Arabidopsis thaliana and we determined that purifying selection has been of great significance in maintaining the function of PKS gene family. From qRT-PCR analysis and a combination of the role of the accumulation of proanthocyanidins (PAs) in brown cotton fibers, we concluded that five PKS genes are candidate genes involved in brown cotton fiber pigment synthesis. These results are important for the further study of brown cotton PKS genes. It not only reveals the relationship between PKS gene family and pigment in brown cotton, but also creates conditions for improving the quality of brown cotton fiber.

Introduction

Plant polyketone compounds are secondary metabolites having a cyclic structure with an oxygen atom bound to the carbon ring. This group includes phenols, stilbene and flavonoid compounds (Abe & Morita, 2010). Owing to the complexity and variety of the pathways and mechanisms of biosynthesis, the number of polyketone compounds is very large and their molecular structures are complex. This complexity results in the compounds having prominent and varied biological activities (Austin & Noel, 2002). The biosynthesis of this group has a common mechanism that includes the enzyme polyketide synthase (PKS). According to the structure of the protein, PKS can be divided into PKS I, II and III (Funa et al., 1999). PKS I and PKS II only exist in microorganisms. Each form has many functional modules and monofunctional subunits (Xie et al., 2016). The PKS III gene family exists mainly in the plant kingdom, but some occur in a few species of bacteria. PKS III gene family members can catalyse plant secondary metabolites having various structures, biological activities and chalcone synthase (CHS) backbones. Examples of such metabolites include chalcone, stilbene, benzophenone, acridone, phloroglucinol, resorcinol and pyrone (Austin & Noel, 2002). These secondary metabolites play important roles in the colouring of plant organs, safeguarding from pesticides and prevention of UV irradiation damage (Li et al., 2016).

The type III PKS gene family is divided into chalcone synthase (CHS) and chalcone synthase-like protein (CHSL) subfamilies. Chalcone synthase is the core enzyme of the PKS III gene family and is the first key enzyme for the plant flavonoid synthesis pathway and the rate-limiting enzyme (Martinez-Perez et al., 2014). The PKS III gene family also includes a series of gene duplications and functional differentiation derived from the class of CHS-like proteins (CHSL) (Eom & Hyun, 2016). CHSL protein is far from the biosynthesis of PAs, The main role is to help plants adapt to changes in the environment, especially in response to fungal invasion (Han et al., 2014). The CHSL of the PKS III gene family include 2-pyrone synthase cloned from Gerbera hybrida (Helariutta et al., 1995), acridone synthase cloned from (Junghanns et al., 1995), benzalacetone synthase cloned from Rheum palmatum (Abe et al., 2001) and stilbene synthase cloned from Pinus sylvestris (Schanz, Schröder & Schröder, 1992). Because of the evolution from a common ancestor, PKS III gene family members have a high degree of homology between the structure and catalytic mechanisms and are very similar. For example, their proteins are essentially homodimers consisting of 40–45 kDa subunits and their active sites have a catalytic triad that is composed of Cys-His-Asn. The functional differences of CHSL and CHS lie in the preference towards different substrates when catalytic reactions occur, changes in the malonyl-CoA number of condensation and different cyclic ways of production (Schröder, 2000).

The first PKS gene was reported in 1983 in a study of PcCHS in Petroselinum crispum and was shown to be involved in the biosynthesis of flavonoids (Reimold et al., 1983). The study of the PKS III gene family continues today. Chalcone synthase (CHS) is by far the most thoroughly studied type III polyketide synthase. CHS catalyses the first step in the synthesis of flavonoids and CHS is responsible for catalysing the reaction of 1 molecule of 4-benzoyl-CoA with 3 molecules of malonyl-CoA to form chalcone (Burbulis & Winkel-Shirley, 1999), the precursor of many flavonoid compounds. The enzymes chalcone isomerase (CHI), flavanone 3-hydroxylase (F3H), flavonoid 3t′-hydroxylase (F3′H), dihydroflavone-4-reductase (DFR) and other enzymes have a common catalytic role in the formation of a variety of flavonoids (Feng et al., 2013). Currently, the cloning and functional analysis of CHS have been reported for many species, e.g., Oryza sativa (Hu et al., 2017), Hypericum monogynum (Jepson et al., 2014), Gerbera hybrida (Helariutta et al., 1995), Petunia hybrida (Koes et al., 1989), Malus domestica (Dare et al., 2013) and Glycine max (Tuteja et al., 2004).

Study of the PKS III gene family in the important cash crop cotton has yet to be conducted. Cotton is an important fiber crop, but it is also used for oil (Cui et al., 2017), drugs (Stipanovic et al., 2005) and other purposes. Naturally colored cotton can be divided into two categories: brown cotton and green cotton. It can synthesize and accumulate pigment to make mature fibers with varied colours during fiber development (Yuan et al., 2012). At present, the application and cultivation of a wide range of naturally coloured cotton varieties produce mainly brown cotton and green cotton. Brown cotton fiber pigments are more stable than those of green cotton; this, combined with its high yield, has led to brown cotton becoming the dominant colour of natural cotton varieties (Qian et al., 2015). Brown cotton is widely favoured for its commercial value and application characteristics, including the lack of need for dyeing, its anti-static electricity properties, ultraviolet resistance and good flame retardance (Hinchliffe et al., 2016). Brown cotton flavonoids are also closely related to resistance to pests and diseases; increasing the flavonoid content can increase plant resistance to insects and thus brown cotton has been widely favoured with increasing commercial value and application prospects (Fan et al., 2016). However, brown cotton fibers do have some problems; these include poor pigment stability, uneven pigment distribution and poor fiber quality (Hua et al., 2007). These problems can restrict the market value of brown cotton. To solve these problems, we focused on the synthesis of brown cotton pigment to improve the quality of brown cotton at the molecular level. At present, many studies have shown that brown cotton pigment is mainly composed of PAs (Gao et al., 2016). In addition, high quality varieties rich in procyanidins are reported in many species, these breeds not only have high commercial value but also help to improve our understanding of flavonoid metabolic pathways precious resources. For example: black rice since ancient times is a very precious ingredient, the color of this grain deepened is due to the accumulation of PAs in rice (Oikawa et al., 2015); Solanum tuberdsm has high intensity of coloring and high nutritional value of food, which are due to Solanum tuberdsm rich in PAs (Gras et al., 2017); what we used to know is that corn are orange particles, but purple corn is more resistant to storage than orange corn and has higher nutritional value (Luna-Vital et al., 2017). These varieties are all rich in PAs, PAs metabolism is an important branch of flavonoid metabolism and thus the PKS III gene family plays an important role in the synthesis of PAs. Thus it can be seen that the study of the PKS gene family is very important not only in brown cotton, but also has a very important significance in many species. The study of PKS gene family can not only help us to better understand the metabolic pathway of flavonoids but also can produce huge commercial value.

Although the whole genome of upland cotton (Gossypium hirsutum) has been sequenced (Li et al., 2015), the whole genome identification and analysis of the type III polyketide synthase family in terrestrial cotton have not yet been reported. The relationship between PKS genes and fiber quality in brown cotton remains unknown. In the present study, we screened the PKS family in upland cotton and analysed the characteristics of its evolution, gene structure, conserved motifs and duplication events. The study species for comparison of the PKS III gene family included Populus tremula, Arabidopsis thaliana, Vitis vinifera and Malus domestica. Arabidopsis thaliana is a widely used research plant and its synthesis of flavonoids is more thoroughly understood, while the other three species are rich in flavonoids. Therefore, the choice of these four species for comparison with the upland cotton can help us better understand terrestrial cotton flavonoid metabolism. According to the analysis of promoter cis-acting elements and the expression patterns of PKS genes in upland cotton, the candidate PKS genes relating to the brown cotton fiber pigment were identified, which provides an important theoretical foundation and genetic resource for improving the uneven distribution, poor stability and fiber quality of natural brown cotton. At the same time, we further analysed the expression patterns of PKS family members and discussed their relationship with the changes in PAs at different developmental stages to determine the PKS candidate genes associated with brown cotton fiber pigment. These results will provide an important theoretical basis for improving the uneven distribution and poor stability of natural brown cotton pigment.

Materials and Methods

Plant materials

Brown cotton plants used line Zongcaixuan No. 1 (brown fiber line) in the experiment were grown in an agricultural park (Hefei, Anhui, China). This brown cotton line belongs to tetraploid upland cotton. In July 2016, 50 brown cotton plants with good growth characteristics were selected at the blooming stage. We began collecting cotton bolls after 3, 6, 9, 12, 15, 18 and 21 days after flowering (DAF). The experimental materials were frozen in liquid nitrogen and quickly transferred to the laboratory refrigerator.

Identification and collection of PKS proteins

In our study, the genomic data of Gossypium hirsutum, Populus tremula, Vitis vinifera, Malus domestica and Arabidopsis thaliana were downloaded from the Phytozome database (Hu et al., 2016) (https://phytozome.jgi.doe.gov/pz/portal.html). DNATOOLS software was used to establish a local database of the amino acid sequences (Curran & Tvedebrink, 2013), including the whole genomes of Gossypium hirsutum, Populus tremula, Vitis vinifera, Malus domestica and Arabidopsis thaliana. The sequences in TBlastN (E-value = 0.001) were queried according to the two conservative domains Chal_sti_synt_N and Chal_sti_synt_C (Han et al., 2016) and compared with the established local database sequences of Gossypium hirsutum, Populus tremula, Vitis vinifera, Malus domestica and Arabidopsis thaliana. Preliminary PKS candidate gene sequences were screened out. The PKS candidate gene sequences obtained by BLAST were tested for whether they contained the two conserved Chal_sti_synt_N and Chal_sti_synt_C domains using Pfam (Bateman et al., 2004) (http://pfam.xfam.org/) and SMART (Letunic, Doerks & Bork, 2012) (http://smart.embl-heidelberg.de/) online software. Multiple sequence alignment and repeat sequence removal were analysed using the ClustalW tool of the MEGA 7.0 software (Kumar, Stecher & Tamura, 2016). The molecular weight of the PKS protein was predicted using the ExPASy Proteomics Server software (Artimo et al., 2012) (http://web.expasy.org/protparam/). WoLFPSORT (Horton et al., 2007) (http://www.genscript.com/wolf-psort.html) was used to predict the PKS protein subcellular localization.

Phylogenetic analysis

Protein sequence alignment was performed using the Clustal X program (Des Higgins, Dublin, Ireland). The phylogenetic tree was built using the Neighbour-Joining (N-J) method with 1,000 bootstraps and MEGA 7.0 (Kumar, Stecher & Tamura, 2016). The GhPKS genes were classified according to the phylogenetic relationships. If two different species of genes are located in the phylogenetic tree at the same node and the sequence similarity is more than 80%, we consider two of these are orthologous genes (Van der Heijden et al., 2007).

Gene structural and conserved motif analysis

The map of the PKS gene structure including Gossypium hirsutum, Populus tremula, Vitis vinifera, Malus domestica and Arabidopsis thaliana was displayed using Gene Structure Server (Guo et al., 2007) (http://gsds.cbi.pku.edu.cn). The motifs of PKS genes in Gossypium hirsutum, Populus tremula, Vitis vinifera, Malus domestica and Arabidopsis thaliana were analysed using MEME online analysis software (Bailey et al., 2015) (http://meme.sdsc.edu/meme4_3_0/intro.html). The specific parameters were as follows: the motif number was 20 and the minimum and maximum widths were 6 and 200, respectively. The motif annotations were obtained from the SMART and Pfam databases.

Chromosomal location and gene duplication

Chromosome starting position and other relevant information concerning the PKS genes were obtained from the public genome database of Gossypium hirsutum, Populus tremula, Vitis vinifera, Malus domestica and Arabidopsis thaliana. The chromosome physical locations of the PKS genes of all five species were obtained using MapInspect (Niu et al., 2016) (http://mapinspect.software.informer.com) software. The gene is located on the same chromosome, separated from the 200 kb and more than 80% similarity gene called tandem duplication; whereas genes that duplicated genes on different chromosomes and more than 80% similarity gene called fragment duplication (Long & Thornton, 2001). Non-synonymous (Ka) and synonymous (Ks) sites were calculated using the DnasP v5.0 software (Librado & Rozas, 2009). Sliding window analysis was also performed using the DnasP v5.0 software; the parameters were as follows: window size, 150 bp; step size, 9 bp.

Upland cotton PKS gene promoter cis-acting element analysis

The promoter sequence of each PKS gene was obtained from the genome database for Gossypium hirsutum, Populus tremula, Vitis vinifera, Malus domestica and Arabidopsis thaliana; this includes the DNA sequence of the initiation codon (ATG) located 1,500 bp upstream of each PKS gene. We used the online software Plantcare (Rombauts et al., 1999) (http://bioinformatics.psb.ugent.be/webtools/plantcarere/html/) to analyse the promoter region cis-acting elements.

RNA extraction and qRT-PCR

In this study, 11 PKS genes of upland cotton were quantitatively analysed by real-time fluorescence. Cotton bolls at 3 DAF, 6 DAF, 9 DAF, 12 DAF, 15 DAF, 18 DAF, 21 DAF were collected and RNA was extracted using the Tiangen (Beijing, China) plant RNA extraction kit. Reverse transcription was performed using a PrimeScript™ RT reagent kit with gDNA Eraser (Takara, Tokyo, Japan) and each reaction used 1 µg of RNA. The specific primers for the PKS gene of upland cotton (Table S1) were designed using Beacon Designer 7 software and the internal reference gene used UBQ7 (Table S1). The qRT-PCR system consisted of 20 µL: 10 µL of SYBR® Premix Ex Taq™ II (2 ×) (Takara), 2 µL of cDNA and 0.8 µL of GhPKS-F and GhPKS-R. The reaction procedure was 40 cycles of 50 °C for 2 min, 95 °C for 30 s, 95 °C for 5 s and 60 °C for 20 s, followed by 72 °C for 10 min; the experiment was repeated three times. Finally, we used 2−ΔΔCt for the calculation of relative expression (Livak & Schmittgen, 2001).

Determination of proanthocyanidin content in brown cotton fibers

The fibers of brown cotton bolls at 3 DAF, 6 DAF, 9 DAF, 12 DAF, 15 DAF, 18 DAF, 21 DAF were stripped, extracted with 80% methanol and subjected to ultrasonic extraction for 30 min. After centrifuging for 15 min, the resulting supernatant was analysed for soluble PAs. A methanol solution containing 1% HCl was added to the precipitate and the solution was placed in a 6 °C water bath for 1 h; after centrifugation for 15 min, the supernatant contained the insoluble PAs. The content of PAs was determined by the method of n-butanol-hydrochloric acid: 400 µL of procyanidin extract was added to 1.5 mL of n-butanol (containing 5% hydrochloric acid) in a boiling water bath for 20 min, after which the absorbance read at 550 nm (Ikegami et al., 2009).

Results

Identification and evolutionary analysis using five genomes

Two kinds of plant PKS III genes conserved domains, Chal_sti_synt_N and Chal_sti_synt_C, were obtained from the Pfam protein database using a hidden Markov model. The two conserved domains have the respective molecular functions of transacylation and transferase. Sequences from TBlastN (E-value = 0.001) were compared to the genome database of Gossypium hirsutum, Populus tremula, Vitis vinifera, Malus domestica and Arabidopsis thaliana using Chal_sti_synt_N and Chal_sti_synt_C. A total of 52 PKS genes were identified (Table S2), including 11 in Gossypium hirsutum (GhPKS1-GhPKS11), 14 in Populus tremula (PtPKS1-PtPKS14), 13 in Vitis vinifera (VvPKS1-VvPKS13), 10 in Malus domestica (MdPKS1-MdPKS10) and 4 in Arabidopsis thaliana (AtPKS1-AtPKS4). In addition to the small number of PKS genes in Arabidopsis thaliana, the number of PKS genes in other species was not very different. To clarify the evolutionary relationships between the 11 PKS genes and PKS genes in four other cultivars, we constructed a phylogenetic tree for a total of 52 PKS genes (Fig. 1). According to the phylogenetic tree nodes, the 52 PKS genes can be divided into four subfamilies: I, II, III and IV. The number of subfamily I members was 17, followed by subfamily IV (12), subfamily III (11) and the lowest number of members was in the subfamily II (10). PtPKS5 and PtPKS7 separated into a class. Among the four subfamilies, the subfamilies I, IV included all five species and each species provided at least one PKS gene. Subfamily III contained three species (Gossypium hirsutum, Populus tremula and Malus domestica), while subfamily II only consisted of Vitis vinifera. It is noteworthy that subfamily I includes an Arabidopsis thaliana PKS gene (AtPKS4) (Owens et al., 2008). This gene is a CHS gene that has been reported in Arabidopsis thaliana. Arabidopsis thaliana plants were treated with high-intensity light for 24 h, resulting in a 50-fold increase in chalcone synthase activity and the accumulation of large amounts of anthocyanins (Courtney-gutterson et al., 1994). The four GhPKSs (GhPKS5, GhPKS9, GhPKS10, GhPKS11) were present in subfamily I, which may indicate that they are closely related to the accumulation of brown cotton fiber pigments. In addition, according to the results of the phylogenetic tree, there were no orthologous genes between the five species.

Figure 1 Phylogenetic analysis of PKS genes in upland cotton (Gossypium hirsutum), Populus tremula, Vitis vinifera, Malus domestica, and Arabidopsis thaliana.

The PKS gene of each species is represented by a different colour: red indicates upland cotton; green represents Populus tremula; purple represents Vitis vinifera; pale blue represents Malus domestica; and the deep blue indicates Arabidopsis thaliana. According to the phylogenetic tree nodes, the PKS genes were divided into four subfamilies (PtPKS5 and PtPKS7 were placed separately into a class). Specific gene names are listed in Table S2.

Figure 2 Exon-intron structure and motif composition of PKS genes across five plant species.

(A) Gene structures of the PKS genes. (B) Distribution of MEME motifs in PKS genes. (C) Gene structure element and motif BOX serial number.

Structural and conserved motif analysis of PKS proteins

To understand the structural diversity of the PKS gene in a more comprehensive way, exon-intron pattern maps were constructed for the 52 PKS genes. As seen from the figure (Fig. 2A), there are 38 members of the 52 PKS genes consisting of two exons and one intron and as in previous reports, most of the plant PKS genes contain two exons and one intron (Durbin, Mccaig & Clegg, 2000). In the remaining 14 members, VvPKS3 contains an exon and an intron. There are six members (GhPKS9, MdPKS3, PtPKS7, VvPKS6, VvPKS8, VvPKS9) with no introns. The remaining seven members (AtPKS3, MdPKS8, VvPKS2, VvPKS4, VvPKS5, VvPKS11 and VvPKS13) are composed of three exons and two introns. VvPKS12 has the largest number with five exons and four introns. There were no UTR regions found in the 23 PKS genes of Gossypium hirsutum and Malus domestica, while 73% of the members of the Populus tremula, Vitis vinifera and Arabidopsis thaliana group had at least one UTR region. The results indicated that the structures of these genes were more complex. All the above results show that the PKS gene family has a diverse genetic structure, which helps to explain the divergence of PKS gene family members. To clarify the structures of the PKS genes, we attempted to gain a better understanding of the conserved motifs of these genes; we thus identified 20 conserved motifs (6–200 amino acid residue widths) using the MEME software (Table S3). The probability of occurrence of motifs 1–10 in upland cotton is more than 65%; we refer to this set as “General Motifs”. The remaining motifs11–20 we refer to as “Specific Motifs” (Fig. 3) (Cao et al., 2016). Among the 20 motifs (Fig. 2B) we found that motifs 1, 3, 5, 7 and 12 encode a Chal_sti_synt_N conservative domain. Motifs2, 4, 6 and 13 encode a Chal_sti_synt_C conservative domain. In upland cotton, in spite of GhPKS3 lacking motifs 6, 7 and GhPKS1 lacking motif 6. Almost all PKS family members included motifs 1, 2, 3, 4, 5, 6 and 7. However, in the other four species, this lack of motifs containing the Chal_sti_synt_N and Chal_sti_synt_C domains is more pronounced. For example, Populus tremula PtPKS4, 8 and 11 lack motif 6; Malus domestica MdPKS8 lacks motifs 3, 5 and 7; in Vitis vinifera motifs 5 and 7 are present in only 3 and 4 members, respectively. In addition, motif 12 did not appear in 42 PKS proteins of Gossypium hirsutum, Populus tremula, Malus domestica and Arabidopsis thaliana, but motif 12 appeared only in two of the PKS proteins of Vitis vinifera (VvPKS5, VvPKS10). The frequency of motif 13 is also very low, with a total of only seven PKS family members. In the phylogenetic tree, the nearest members of each subfamily have similar motif combinations. Example combinations include MdPKS7, 9, VvPKS6, 8 and PtPKS4, 11. In addition, there are some proteins belonging to a subfamily with unique motifs. For example motif 15 is unique to subfamily IV and motif 17 only appears in the subfamily III. These subfamily-specific motifs play a very important role in the subfamily PKS proteins regarding function.

Figure 3 Distribution of motifs in PKS proteins from Gossypium hirsutum, Populus tremula, Vitis vinifera, Malus domestica and Arabidopsis thaliana.

Colour key: the depth of colour indicates the percentage of motifs in the species.

Figure 4 Sequence alignment of GhPKSs against the other plant species.

The first line represents the secondary structure of Oryza sativa CHS. The blue box and the red font in the figure represent the conservative amino acid residues, and the sequence of the red regions shows a very high degree of conservation. The black wavy lines and arrows represent α-helices and β-sheet, respectively. The purple five-pointed star represents the catalytic triad, and the active amino acids are expressed in green or black triangles. OsCHS, Oryza sativa chalcone synthase (4350636); AtCHS, Arabidopsis thaliana chalcone synthase (AAB35812.1); MsCHS, Medicago sativa chalcone synthase (P30074).

Comparison of GhPKS protein sequences with those of other plants

We identified and compared 11 sequences of PKS protein in upland cotton with the sequences of Oryza sativa chalcone synthase (OsCHS), Arabidopsis thaliana chalcone synthase (AtCHS) and Medicago sativa chalcone synthase (MsCHS) to clarify the functional divergence of PKS III gene family members. The results are shown in the figure (Fig. 4). The blue box and the red font in the figure represent the conservative amino acid residues and the sequence of the red regions shows a very high degree of conservation. The black wavy lines and arrows represent the α-helix and the β-sheet, respectively. The purple five-pointed star represents the catalytic triad (Cys-His-Asn) and the active amino acids (Thr, Phe, Gly, Ser) in the catalytically active central cavity are expressed as green or black triangles. When the plant PKS III enzyme catalyses the polyketone reaction, the starting substrate is first bound at the Cys in the catalytic triplet, followed by decarboxylation of the malonyl-CoA and the occurrence of the substrate condensation reaction so that the polyketone chain is continuously extended (Jez, Bowman & Noel, 2002). The final intermediate product undergoes a series of complex cyclization reactions that ultimately form the final product (Abe et al., 2001). Active amino acids located in the catalytically active central chamber can adjust the type of reaction-starting substrate and the length of the polyketone chain by adjusting the size of the catalytically active central chamber space (Jez, Bowman & Noel, 2002). The Cys-His-Asn catalysed triplets inherited from keto acyl synthase III (KASIII) (Austin & Noel, 2002) are highly conserved in each sequence in 11 PKS proteins of upland cotton. However, more amino acid substitutions occur at the four active amino acid positions. Thr at GhPKS2, 3, 4, 7 is replaced by a Met. Ser at GhPKS1 is replaced by Lys and in GhPKS6, 8 is replaced by Met at the same position. The active amino acid Phe has two sites in the catalytically active central cavity and is closely related to the decarboxylation reaction of malonyl-CoA, which is represented by a black triangle in the figure. The first Phe active site was highly conserved in all upland cotton PKS proteins, but at the second Phe active site, Phe at GhPKS2, 3, 4, 7 was replaced by Tyr. The active amino acids Thr, Gly and Ser can regulate the specificity of the reaction substrate as well as the product. In the upland cotton PKS protein, amino acid substitution occurs in active amino acids Thr, Gly and Ser in multiple protein sequences; this phenomenon may be closely related to PKS III gene family functional diversity.

Chromosomal localization and gene duplication

To identify the distribution of PKS genes on the chromosome of each species and in the gene cluster, simultaneously to confirm the type of gene duplication events in upland cotton. We mapped the 52 PKS genes in five species (Gossypium hirsutum, Populus tremula, Vitis vinifera, Malus domestica and Arabidopsis thaliana) to identify the chromosomal distribution of these PKS genes (Fig. S1). In our study, the PKS genes in the other four species were unevenly distributed on the chromosomes except for the distribution of the PKS gene in the Vitis vinifera, which was more concentrated on chromosome 16. In upland cotton, the PKS gene distribution was A2_chr6 (1), A2_chr8 (2), A2_chr9 (1), At_chr11 (1), Dt_chr8 (1), Dt_chr10 (1) and Dt_chr11 (4). In Populus tremula, the 14 PKS genes were distributed on chromosomes 1, 2, 3, 4, 5, 9 and 12. In Malus domestica, we found that the PKS genes were distributed on chromosomes 2, 9, 14, 15 and that MdPKS1 was not mapped to any chromosome. The PKS genes in Arabidopsis thaliana are distributed on chromosomes 1, 4 and 5. However, in Vitis vinifera, 10 PKS genes were distributed on chromosome 16 and the remaining 3 PKS genes were distributed on chromosomes 3, 14 and 15. In the evolution of genes, most gene family expansion is due to the phenomenon of gene duplication, including tandem duplication and fragment duplication. To clarify how the PKS gene family was amplified, we examined the duplication of the PKS genes in five species (Gossypium hirsutum, Populus tremula, Vitis vinifera, Malus domestica and Arabidopsis thaliana). Among the 52 PKS genes, we identified 10 gene duplication events in Gossypium hirsutum (2), Populus tremula (3), Vitis vinifera (3) and Malus domestica (2); in Arabidopsis thaliana, no gene duplication events were found. Five pairs of duplicated genes belonged to tandem duplication and five pairs of duplicated genes belonged to fragment duplication (Table 1). After analysing the gene duplication events of the PKS III gene family in five species, we calculated the Ka, Ks and Ka/Ks ratios of the eleven gene duplication events to explore the effects of these genes on the evolutionary processes (Table 1). In general, Ka/Ks < 1 represents negative selection or purification selection, Ka/Ks > 1 represents positive selection and Ka/Ks = 1 indicates neutral selection (Bitocchi et al., 2017). In our study, the Ka/Ks values of the 10 pairs of duplicated genes were less than 0.309. The results indicated that in these five species, the PKS III gene family was expanded due to gene duplication events and these repeated genes that undergo gene duplication experience strong purifying selection. Sometimes positive selection may be masked by strong negative selection. To identify positive selection of PKS loci in the occurrence of gene duplication events, we also performed a sliding window analysis of two pairs of duplicated genes in upland cotton (Fig. S2). There was never more than one repeat locus in the upland cotton, indicating that there was no positive selection for the two pairs of duplicated genes.

Table 1 Ka/Ks analysis for the duplicated PKS paralogues from upland cotton, Populus tremula, Vitis vinifera, and Malus domestica.

The chromosomal localization results are shown in Fig. S1, and the sliding window analysis results are shown in Fig. S2.

Duplicated pairs	Ka	Ks	Ka/Ks	Purifying selection	Duplicated type	
GhPKS5-GhPKS11	0.0159	0.9533	0.017	Yes	Segmental	
GhPKS6-GhPKS8	0.0033	0.0601	0.055	Yes	Segmental	
PtPKS6-PtPKS8	0.0387	0.3075	0.126	Yes	Segmental	
PtPKS4-PtPKS11	0.0475	0.3185	0.149	Yes	Segmental	
PtPKS12-PtPKS13	0.0081	0.1357	0.060	Yes	Tandem	
MdPKS2-MdPKS6	0.009	0.0291	0.309	Yes	Segmental	
MdPKS7-MdPKS9	0.0068	0.3129	0.022	Yes	Tandem	
VvPKS1-VvPKS4	0.0807	0.4019	0.201	Yes	Tandem	
VvPKS6-VvPKS8	0.0094	0.0699	0.134	Yes	Tandem	
VvPKS7-VvPKS9	0.0053	0.0213	0.249	Yes	Tandem	

Analysis of cis-acting elements in the promoter of the PKS gene in upland cotton

To clarify the characteristics of the promoters of PKS genes in upland cotton, we analysed the cis-acting elements of 11 PKS gene promoters in upland cotton (promoter length = 1,500 bp) (Table S4). Strong light can regulate the expression of the PKS gene and there are many cis-acting elements in the promoter regions of PKS genes in upland cotton, e.g., Box4 (ATTAAT), SP1 (CC(G/A)CCC), CATT-Motifs (GCATTC) and many G-Boxes (CACGTT). It has been reported that Arabidopsis thaliana CHS genes were regulated by MYB transcription factors (Chezem & Clay, 2016). In our study, cis-acting elements associated with MYB transcription factors were also found in the promoter region of PKS genes in upland cotton, e.g., MBS (CGGTCA) and MRE (AACCTAA). This suggests that the expression of PKS genes may be regulated by MYB transcription factors. In addition, there are some cis-acting elements related to various life activities, TC-rich repeats (GTTTTCTTAC) associated with defence and stress, anaerobic induction of ARE (TGGTTT) and CGTCA-motifs (CGTCA) related to methyl jasmonate reactions. The specific cis-acting elements, concrete sequences and functions are shown in Table S5.

Expression characteristics of the PKS gene in upland cotton

The function and expression patterns of genes are closely related (Zhang et al., 2014). To explore the expression patterns of PKS genes in upland cotton, we studied the expression patterns of 11 PKS genes in upland cotton at different stages of cotton fiber development, including 3 DAF, 6 DAF, 12 DAF, 15 DAF, 9 DAF, 18 DAF, 21 DAF and different plant parts, including roots, stems, leaves, fiber (cotton fiber development represented by 6 DAF). GhPKS8 is a special gene in the 11 upland cotton PKS genes because no expression was detected in any tissue at any stage of cotton fiber development. The other 10 PKS genes were detected in all tissues and at different stages of cotton fiber development (Fig. 5). We found that GhPKS1 is present at a higher level of transcription in the roots. GhPKS2, 3 and 7 showed a high expression level in the stems, while the expression levels in roots, leaves and fiber were low. GhPKS4 showed high levels of transcription in all tissues of upland cotton. GhPKS6 was highly expressed in the leaves, while the expression of GhPKS5, 9, 10 and 11 in cotton fiber was significantly higher than that in the other three plant tissues. The results of expression patterns of the 11 PKS genes in different tissues of upland cotton showed that GhPKS5, 9, 10 and 11 were mainly expressed in upland cotton fibers. We analysed the expression patterns of 11 PKS genes in upland cotton at different stages of cotton fiber development. The results showed that 11 PKS genes had multiple expression patterns. GhPKS1, 6 and 10 showed a gradual increase in transcription level from 3 DAF–15 DAF and the transcriptional level began to decrease after 15 DAF. GhPKS2, 7 had higher transcription levels at the later stages of fiber development and GhPKS3, 9 reached their highest levels at 12 DAF. GhPKS4, 5, 9 and 11 showed the highest amounts of transcriptional accumulation in the early stages of cotton fiber development. In brown cotton fibers, PAs are the main precursors of pigment. We also studied the accumulation of PAs in the fibers of brown cotton at different developmental stages (Fig. 6). The determination of PAs showed that both soluble and insoluble PAs had mainly accumulated before 15 DAF, after which its content gradually decreased; these results were consistent with the previously reported results (Li et al., 2012). Interestingly, GhPKS4, 5, 9, 11 had a higher level of transcription at the early stages of cotton fiber development; the amount of expression then decreased gradually, which is consistent with the rule of accumulation of PAs in brown cotton fibers.

Figure 5 Expression patterns of PKS genes of upland cotton in different tissues and brown cotton fibers at different growth stages.

(A–J) Expression patterns of PKS genes in upland cotton in different tissues. (K–T) Expression patterns of PKS genes in upland cotton at different growth stages of cotton fibers.

Discussion

The plant PKS III gene family, which only exists in the plant kingdom, is associated with a variety of plant life activities (Shimizu, Ogata & Goto, 2017). The PKS III gene family is not very large and PKS III gene family members have been identified or cloned in several species. For instance, 14 PKS genes have been identified in Zea mays (Han et al., 2016), 12 PKS genes have been isolated and sequenced in Petunia hybrida (Koes et al., 1989) and 27 PKS genes have been reported in Oryza sativa (Hu et al., 2017), which is the species with the largest number of PKS genes reported to date. In this study, we identified 11 PKS genes from upland cotton and compared these with PKS genes in Populus tremula (14), Vitis vinifera (13), Malus domestica (10) and Arabidopsis thaliana (4). The 52 PKS genes were divided into four subfamilies, I, II, III and IV, according to the phylogenetic tree nodes. Previous researchers have suggested that most of the CHS genes consist of two exons and one intron (Durbin, Mccaig & Clegg, 2000) and the diversity of gene structures is important for the evolution of gene families (Swarbreck et al., 2008). According to our study, 72% of the 52 PKS genes consisted of two exons and one intron. However, some genes also had different compositions. For example, VvPKS12 consists of five exons and four introns. Six PKS genes including GhPKS9 had no introns and seven PKS genes had three exons and two introns.

Figure 6 The content of PAs at different fiber development stages in brown cotton.

The contents of soluble proanthocyanidins, insoluble proanthocyanidins and total proanthocyanidins are expressed as different colours.

We identified 20 conservative motifs using MEME software (Bailey et al., 2015). Among these 20 motifs, motifs 1, 3, 5, 7 encoded a Chal_sti_synt_N conservative domain and motifs 2, 4, 6, 13 encoded a Chal_sti_synt_C conservative domain. All 52 PKS genes with motifs encoding these two conserved domains demonstrate that the PKS III gene family has been highly conserved during evolution. These two conserved domains are associated respectively with acyl transfer activity and transferase activity (Götz et al., 2008), which indicates that these genes function in catalysing the formation of polyketone compounds. We found that the PKS genes in the same subfamily had similar motif compositions, e.g., MdPKS7, 9, VvPKS6, 8 and PtPKS4, 11. At the same time, there were some subfamily-specific motifs. The diversity of gene structure and conserved motif distribution may help to explain the functional dispersion of PKS gene family members.

The plant PKS III enzyme protein-specific catalytic triad composed of Cys-His-Asn could be traced back to the earliest ancestors of KAS III (Austin & Noel, 2002), which was considered to be important for the maintenance of PKS III gene family functions. Therefore, using BLAST, we queried the protein sequences of the 11 upland cotton PKS genes and the AtCHS and MsCHS protein sequences with the reported secondary structure of OsCHS as a template (Consortium et al., 2003). The results showed that the Cys-His-Asn catalytic triad was highly conserved in all GhPKS sequences. However, there were more amino acid substitutions in active amino acids in the catalytic active site. For example, the first Phe site was highly conserved in all GhPKSs in the two Phe sites that are closely related to the binding of various CoA, while more amino acid substitutions appear in the second Phe site. At the same time, the three active amino acids (Thr, Gly, Ser), which are responsible for the regulation of the substrate and the length of the polyketide chain, have also been replaced by other amino acids. This suggested that the catalytic triad of the GhPKS protein was highly conserved in the process of gene evolution, whereas the active amino acids were not highly conserved. Therefore, we speculated that the diversity of amino acids at the active amino acid sites was the main cause of the functional dispersion of the PKS gene family.

Chromosomal localization analysis showed that the distribution of PKS genes in five species in our study was irregular. The PKS genes were more concentrated on chromosome 16 except for the PKS genes in Vitis vinifera. The rest of the PKS genes were scattered on multiple chromosomes, which is consistent with previous studies (Han et al., 2016). Subsequently, we found 10 pairs of duplicated genes in the five species: two pairs in upland cotton, three pairs in Populus tremula, three pairs in Vitis vinifera and two pairs in Malus domestica. No duplicated genes were found in Arabidopsis thaliana. Among the 10 pairs of duplicated genes, five of the duplicated genes in the Vitis vinifera, Malus domestica and Populus tremula were from tandem duplication and the other five pairs of duplicated genes were derived from segmental duplication. It has been reported that there are seven pairs of duplicated genes in the 27 PKS genes of Oryza sativa, but only one pair of duplicated genes was the result of segmental duplication (Hu et al., 2017). In Zea mays, there were two pairs of duplicated genes in the 14 PKS genes and these were from segmental duplication (Han et al., 2016). The PKS gene family in Oryza sativa has many duplicated genes and there are two types of gene duplication in Oryza sativa, tandem duplication and fragment duplication, which also explains why the number of PKS genes in Oryza sativa is greater than that of other species. We speculated that there were two kinds of duplication modes in the process of PKS gene duplication in terrestrial plants: tandem duplication and fragment duplication. However, it is unknown whether the duplications were mainly in the form of tandem duplication or segmental duplication, which have varied tendencies in different plants. It is generally believed that tandem duplication contributes to the generation of new genes and fragment duplication leads to slower evolution of the gene family (Cao et al., 2016). In upland cotton, the duplications of the PKS gene carried out in the form of segmental duplication indicated that the evolution of the PKS gene family was slow. The analysis of the Ka/Ks values of the 10 repeat genes showed that the Ka/Ks values of the 10 duplicated gene pairs were less than 0.309, which indicated that these replicates had undergone strong purification selection after duplication was complete. This was for a factor in maintaining the PKS gene family.

In Arabidopsis thaliana, AtCHS is regulated by a variety of MYB transcription factors such as AtMYB11, 58, 63, 111 and other transcription factors that can activate AtCHS transcription (Chezem & Clay, 2016). Furthermore, Arabidopsis thaliana treated with high-intensity light for 24 h resulted in a 50-fold increase in the activity of chalcone synthase and a large amount of anthocyanin accumulation (Courtney-gutterson et al., 1994). In this study, the analysis of the cis-elements in the promoter regions of these 11 PKS genes of upland cotton showed that the regions contained many elements related to light regulation and MYB transcription factor binding. Therefore, we believe that upland cotton PKS genes may be regulated by light and MYB transcription factors. The expression patterns of PKS genes of upland cotton in different tissues and cotton fiber development were studied by qRT-PCR. GhPKS1 showed a higher transcription level in the roots; GhPKS2, 3, 7 showed a high expression level in the stem; GhPKS5, 9, 10, 11 were mainly expressed in the fibers. The accumulation of PAs in brown cotton fibers occurred mainly before stage 15 DAF of cotton fiber development (Li et al., 2012). The expression of GhPKS4, 5, 9, 11 was higher in the early stages of cotton fiber development and PAs in the brown cotton fibers gradually accumulated as their expression increased. The procyanidin content then decreased as the amount of expression also gradually decreased. Previous studies have shown that the PKS gene encodes a key enzyme in the flavonoid biosynthetic pathway as the first rate-limiting enzyme (Martinez-Perez et al., 2014). The precursor material of the pigment in the brown cotton fiber is PAs, which are flavonoids (Liu et al., 2016). The expression trend of GhPKS4, 5, 9, 11 was consistent with the trend of the accumulation of PAs in brown cotton fibers; therefore, we speculate that GhPKS4, 5, 9, 11 may be involved in brown cotton fiber pigment biosynthesis.

Conclusion

In this study, we identified 11 PKS genes from upland cotton and compared them with analogous genes from Populus tremula, Arabidopsis thaliana, Vitis vinifera and Malus domestica; there were 41 PKS genes with respect to phylogeny, gene structure, conserved motifs and selection pressure. According to the constructed phylogenetic tree, the 52 total PKS genes were divided into four subfamilies. Most of the PKS genes were composed of two exons and one intron. The PKS genes in the same subfamily had similar gene structure and conserved motifs. At the same time, our research on structure showed that gene duplication has been the main driving force of the expansion of the PKS III gene family, but there is a kind of species-specificity concerning fragment duplication vs. tandem duplication. The results of the Ka/Ks ratio analysis showed that purification selection has been important in maintaining the function of the PKS III gene family. According to the analysis of cis-acting elements of PKS promoters in upland cotton, the PKS gene may be regulated by MYB transcription factors and light. The analysis of qRT-PCR and the accumulation of PAs in brown cotton fibers suggest that GhPKS4, 5, 9 and 11 may be involved in the accumulation of PAs in brown cotton fibers.

Supplemental Information

Table S1 Primers used in RT-PCR

Click here for additional data file.

Table S2 The PKS genes identified in this study are listed

Click here for additional data file.

Table S3 Detailed information of the 20 motifs in the 52 PKS proteins

Click here for additional data file.

Table S4 Analysis of cis-acting elements of PKS gene promoter in upland cotton

Click here for additional data file.

Table S5 Potential cis-elements in the 5′ regulatory sequences of the 11 GhPKS genes

Click here for additional data file.

Figure S1 Chromosomal locations of PKS genes in five species

(A) Chromosome localization of 14 PKS genes in Populus tremula. (B) Chromosome localization of 10 PKS genes in Malus domestica. (C) Chromosome localization of 4 PKS genes in Arabidopsis thaliana. (D) Chromosome localization of 10 PKS genes in Vitis vinifera. (E) Chromosome localization of 11 PKS genes in Gossypium hirsutum.

Click here for additional data file.

Figure S2 Sliding window analysis of 2 pairs of duplicated genes in upland cotton

Click here for additional data file.

The authors are deeply grateful to Prof. Yongping Cai and Prof. Yi Lin, who provided the sample used in the study and very effective direction. The authors also thank Prof. JunShan Gao, Prof. Dahui Li, Dr. Yanan Wang, Dr. Xi Cheng and Dr. Muhammad Abdullah for providing valuable suggestions and comments.

Additional Information and Declarations

Competing Interests

Author Contributions

Data Availability

The authors declare there are no competing interests.

Xueqiang Su conceived and designed the experiments, performed the experiments, analyzed the data, contributed reagents/materials/analysis tools, wrote the paper, prepared figures and/or tables.

Xu Sun performed the experiments, analyzed the data, contributed reagents/materials/analysis tools, prepared figures and/or tables.

Xi Cheng performed the experiments, wrote the paper, prepared figures and/or tables.

Yanan Wang performed the experiments, prepared figures and/or tables.

Muhammad Abdullah wrote the paper.

Manli Li performed the experiments, wrote the paper.

Dahui Li and Junshan Gao performed the experiments, reviewed drafts of the paper.

Yongping Cai and Yi Lin conceived and designed the experiments, reviewed drafts of the paper.

The following information was supplied regarding data availability:

The raw data has been provided as a Supplemental File. The following information is available at https://phytozome.jgi.doe.gov/pz/portal.html: The Gossypium hirsutum genome (version1_0.fa), annotation information (version1_0.gff), coding sequences (version1_0.cds.fa) and protein sequences (version1_0.pep.fa); the Populus tremula protein sequences (version3_0.pep.fa); annotation information (version3_0.gene.gff3); the Vitis vinifera protein sequences (version12x.pep.fa), annotation information (version12X.gene.gff3); the Malus domestica protein sequences (version1_0.pep.fa), annotation information (version1_0.gene.gff3); the Arabidopsis thaliana protein sequences (version167_TAIR10.pep.fa), annotation information (version167_TAIR10.gene.gff3).

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
