# Peer review of "Comparative genomic analysis of the PKS genes in five species and expression analysis in upland cotton"

_PeerJ, doi:10.7717/peerj.3974_

## Round 0.1 · original submission · Major Revisions

You must have a native English speaker correct all the mistakes, as outlined by the reviewers. Also please answer the other comments and increase the legibility of the figures. In addition, there are many many sequenced plant genomes, so you need to better justify why you only compared to a few, or increase the number of genomes used in your analysis.

Reviewer 1 ·

Basic reporting

The MS (##18119) reported the PKS genes and its expression profile in upland cotton. The results are overall interesting and could be published. However, the MS is poor prepared. Please see my comments and suggestions below!

Overall, I think there are three major problems should be addressed.
1. The PKS genes were found in many plants. why are the PKS genes selected from these species (Populus tremula, Vitis vinifera, Malus domestica, and Arabidopsis thaliana) compared with cotton PKS genes?
2. Most figures are too small to see clearly.
3. There are many mistakes in the MS, including abbreviations, spaces between words, singular and plural, etc. The English language should be improved. I suggest that you have a native English speaker review your manuscript.

Experimental design

no comment

Validity of the findings

no comment

·

Basic reporting

The manuscript needs extensive editing to improve its language and grammar. For example, the first abstract seems to be less well edited and states “Plant type III polyketide synthase can catalyze into a series of secondary metabolites …”. I believe the authors meant to write that PKS can catalyze reactions forming secondary metabolites. Line 25 should read “ … candidate genes involved in brown cotton fiber synthesis …” rather than ““ … candidate genes involving in brown cotton fiber synthesis …”. I think that the column heading in Table S2 should read “No. of exons” rather than “No. of extrons”. There are other sentences throughout the manuscript that need to be edited.

The introduction should lead the reader to understand why the problem is important and why certain genes are studied. The article aims to study the plant polyketide synthase genes in upland cotton and compare them to other plant species. This is a valid and appropriate question but the introduction could be improved to state any specific and additional reasons as to why this question is important. For example, is the pigmentation of brown cotton fibers of great commercial importance because it commands a higher price? Is the study of PKS genes in upland cotton also important because PKS genes might be implicated in stress response and tolerance, thus helping to produce sturdier versions of a very widely grown crop?

All acronyms should first be introduced with their full names and then used thereafter. For example, CHS is first used in line 41, and the reader is only introduced to the full term in line 60. Also, it is initially unclear why CHS is discussed until about lines 70-80 where it is explained that CHS is an important PKS gene involved in many processes including the regulation of pigmentation.

All previous results and data mentioned in the manuscript should refer to the appropriate publications. For example, line 83 mentions that the upland cotton genome has been sequenced and assembled but no citation is given. In addition, appropriate scientific names/accession names of the upland cotton and brown cotton lines.

All the software tools mentioned such as DNATOOLS, MEGA, Plantcare etc. should also be accompanied by the citations of any respective publications. For some of the previous results mentioned, it is not clear which publication is the relevant citation. For example, two papers are simultaneously cited in line 197, but it is not clear whether they both mention the increase in chalcone synthase activity or if one of them is about the four PKS genes mentioned next.

Some of the figures are too small to be legible. Specifically, Figures 4, 5, and S1 are not legible in the current resolution and size.

Line 342 mentions Li T 2012 but this publication is not listed in the references.
On line 411, Cao et al., 2016 is cited for the discussion of comparative tandem duplication and fragment duplication evolution rates. Is this the first or most prominent publication finding this result?

Figures 5 and 6 represent the experimental data generated in this manuscript, and the raw data for these figures has been provided.

Experimental design

I do not have any major objection to the scientific validity of the methodology of the experiments and the bioinformatics analyses carried out by the authors. This review is not supposed to judge the impact or novelty of the results and findings. However, it will be very useful to the reader that the results presented here are discussed to illuminate the broader scientific questions they answer or raise. For example, line 234 makes a broad statement that the different motifs found in the different sub-families may be important for the different functional roles performed by the proteins. This seems like a truism, and I suggest the authors try to discuss whether anything is known about the functional role of the motifs they find in the PKS genes studied. Currently, all that the manuscript mentions is that the motif annotations were taken from SMART and Pfam databases. The division into general and specific motifs is due to their abundance in the genes studied, but Cao YP, 2016. I believe the same paper is elsewhere cited as Cao et al., 2016. Is this the same paper? Why is this relevant to the differentiation between general and specific motifs?

The method used to differentiate between tandem and fragment duplication should be clearly mentioned.

The authors discuss the relevance of the Phe sites in the PKS genes, and line 258-260 mentions that these sites are related to the decarboxylation reaction of malonyl CoA. Please provide a reference for this statement. Subsequently, it is mentioned that the second Phe is substituted by Tyr in certain genes and Thr, Gly, and Ser are also replaced in some genes. For these and other substitution sites, is it possible to differentiate between highly conserved sites, background drift, and positive selection? Can the authors calculate site-specific substitution rates or use some other appropriate method to comment on this?

Validity of the findings

As mentioned earlier, I would strongly suggest the authors to discuss the broader scientific context of the specific and quantitative results presented. For example, what is the importance of studying the chromosomal locations of the PKS genes? Is this used to identify the genes as the result of tandem and fragment duplication?

I do not have any major objection to the data quality and the conclusions drawn.

Additional comments

Overall, the manuscript uses appropriate experimental and analytical tools for the questions it studies. The study of genes involved in pigmentation in brown cotton fibers and the PKS genes in upland cotton have scientific and commercial importance.

The manuscript needs major editing for language and presentation. In addition to correcting grammar, some paragraphs, especially in the introduction should be rearranged for better logical flow. This will help the reader and also allow for a better peer review once any ambiguities and distracting language problems are removed.

I have suggested one additional analysis: assigning the site substitutions discussed in the manuscript as conservatory, neutral or positively selected. If the authors feel this should not be done, please mention why it is so.

Reviewer 3 ·

Basic reporting

The paper has a clear structure, sufficient field background, and professional figures and tables.

1. The English writing needs to improve. For example,

line 222-224, "in addition to GhPKS3 lack of motifs6,7 and GhPKS1 lack of motifs6 almost all PKS family members include motifs1,2,3,4,5,6,7 these 7 motifs." in addition to sth -> in spite of sth.
line 224-225, two predicates in "this is less Chal_sti_synt_N and Chal_sti_synt_C domain part of the phenomenon is very obvious."
line 272-273, no predicate in "In upland cotton, PKS gene distribution in A2_chr6
273 (1), A2_chr8 (2), A2_chr9 (1), At_chr11 (1), Dt_chr8 (1), Dt_chr10 (1), Dt_chr11 (4)."

2. For some unknown reason, in many sentences, two individual words are concatenated without spaces in between, making the sentences difficult to understand. For example,

line 37, PKSIand PKSIIonly -> PKS I and PKS II only
line 41, for examplechalcone -> for example, chalcone,
line 110, domainswas -> domains was
line 116, weightof -> weight of
line 113, thalianawere -> thaliana were
line 139, databaseof -> database of
line 187, anddivided -> and divided
line 307, geneswere -> genes were
line 318, functionand -> function and
line 348, onlyexists -> only exists

3. Some abbreviations are not defined. For example,

line 47, CHSL is not defined.
line 81, Proanthocyanidins should occur before PA.

Experimental design

The research fills the knowledge gap of the PKS genes in upland cotton. Some analysis needs to describe in details.

1. For example, it is not clear how did the authors derived the conclusion that no direct homologous genes between the species in line 200-201. Is there any p-value or hypothesis testing that support this conclusion?

2. The authors claimed that "gene duplication is the main driving force of the amplification of PKS III gene family" in line 447-449. However, it is not clear how the conclusion was derived based on the facts that (1) "10 gene replication events were identified in the 52 genes" in line 283, and (2) Ka/Ks<0.4 indicates negative selection in line 292. Is 10/52 significantly high? How is negative selection associated with amplification of PKS gene?

Validity of the findings

In this paper, the authors identified 11 PKS genes in the genomes of upland cotton species. The 11 PKS genes in cotton were compared with PKS genes in other plants.
The expression of the PKS genes was quantitatively measured at various growth stages and tissues. The data is solid and the analysis is sound.

---

## Round 0.2 · Minor Revisions

Please make the final corrections that the reviewer requested.

·

Basic reporting

The manuscript has been edited and the language has improved greatly. Line 19 reads “(a)ccording to phylogenetic tree, a total of 52 PKS genes …”, which should be “according to the phylogenetic tree, a total of 52 PKS genes …” or “according to the constructed phylogenetic tree, a total of 52 PKS genes …”.

The supplementary tables S1-S5 reported in the Excel file do not have column headers.

On line 333 it says“(f)our pairs of duplicated genes belonged to tandem duplication and seven pairs of duplicated genes belonged to fragment duplication”. I understand that these genes can be identified from the figure showing the chromosomal locations of the genes, but it would be helpful if a table lists which genes result from tandem duplication and which genes result from fragment duplication.

Experimental design

Concerns raised in the review have been addressed.

Validity of the findings

Concerns raised in the review have been addressed.

---

## Round 0.3 · accepted · Accept

Your manuscript is now accepted.